# The Effect of Therapeutic Exercise on Body Weight Distribution, Balance, and Stifle Function in Dogs following Stifle Injury

**DOI:** 10.3390/ani14010092

**Published:** 2023-12-27

**Authors:** Ida Brantberg, Wilhelmus J. A. Grooten, Ann Essner

**Affiliations:** 1School of Veterinary Science, The University of Liverpool, Leahurst Campus, Chester High Road, Neston CH64 7TE, Wirral, UK; 2Djursjukhuset Malmö, IVC Evidensia, Cypressvägen 11, SE-213 63 Malmö, Sweden; 3Division of Physiotherapy, Department of Neurobiology, Care Sciences and Society, Karolinska Institutet, Alfred Nobels Allé 23, SE-141 83 Huddinge, Sweden; wim.grooten@ki.se; 4Women’s Health and Allied Health Professionals’ Theme, Department of Occupational Therapy and Physiotherapy, Karolinska University Hospital, Solna N1:00, SE-171 76 Stockholm, Sweden; 5Djurkliniken Gefle, IVC Evidensia, Norra Gatan 1, SE-803 21 Gävle, Sweden; ann.essner@evidensia.se; 6Department of Women’s and Children’s Health, Uppsala University Hospital, SE-751 85 Uppsala, Sweden

**Keywords:** dogs, stifle injury, physiotherapy, therapeutic exercise, balance, stability, weight bearing, rehabilitation

## Abstract

**Simple Summary:**

Stifle injury is common in the companion dog population, often leading to reduced weight bearing due to pain or joint instability, affecting neuromuscular control, balance, and proprioception. Earlier studies have reported the benefits of rehabilitation after stifle injury, though the specific effects of most exercises are unknown and predicting outcomes is challenging. In this randomized and controlled study, we investigated the effects of a 12-week progressive therapeutic home exercise protocol on static body weight distribution between hindlimbs, balance control, pain-related disability, and stifle function in 33 dogs diagnosed with stifle injury after 12 weeks of home exercise. An intervention and a control group both followed a standard rehabilitation protocol for stifle injury, with the intervention group also following a progressive therapeutic exercise protocol. The results indicate that the progressive therapeutic exercise protocol improved static body weight distribution between hindlimbs, pain-related functional disability, and stifle function. The effects were statistically significant, with intermediate to strong effects. There was a small but nonstatistically significant positive effect on balance control with the therapeutic exercise protocol, perhaps due to the rather small sample group and unvalidated method of measurement. We suggest studying balance control in dogs with stifle injuries more thoroughly in future studies.

**Abstract:**

Stifle injury is common in the companion dog population, affecting weight bearing, neuromuscular control, and balance. Therapeutic exercises after stifle injury seem to be effective, but high-quality research evaluating the effects is lacking. This randomized controlled trial evaluated the effects of a 12-week progressive therapeutic home exercise protocol on three-legged standing, targeting balance and postural- and neuromuscular control and disability in dogs with stifle injury. Thirty-three dogs with stifle injury were randomly allocated to intervention (*n* = 18) and control groups (*n* = 15), both receiving a standard rehabilitation protocol. Additionally, the intervention group received a progressive therapeutic exercise protocol. The outcome measures were static body weight distribution between hindlimbs, balance control, the canine brief pain inventory, and the Finnish canine stifle index. Both groups improved after the intervention period, but the group using the progressive therapeutic exercise protocol improved to a greater extent regarding static body weight distribution between the hindlimbs (I: median = 2.5%, IQR = 1.0–4.5; C: median = 5.5%, IQR = 3.0–8.8), pain-related functional disability (I: median = 0.0, IQR = 0.0–0.2; C: median = 0.9, IQR = 0.1–1.8), and stifle function (I: median = 25.0, IQR = 9.4–40.6; C: median = 75.0, IQR = 31.3–87.5), with intermediate to strong effects. These clinically relevant results indicate that this home exercise program can improve hindlimb function and restore neuromuscular control.

## 1. Introduction

Stifle injury is very common in the companion dog population, with cranial cruciate ligament (CCL) injuries and patellar luxation (PL) being the most common orthopedic disorders of the stifle joint [1,2,3]. Insurance data from Sweden showed the overall incidence of stifle injury to be 55.4 cases per 10,000 dog years at risk, with 43.5% affected by CCL injury and 29.8% PL [2]. The CCL and muscles of the quadriceps, biceps femoris, and gastrocnemius require coordination and exact timing to deliver active stifle stability during dynamic movement [4]. Reduced timing or co-contraction may increase the susceptibility of the periarticular structures to injury [5]. Hence, improving the motor control mechanism is anticipated to increase dynamic joint stability, protecting the CCL from excessive strain [5,6].

Surgical treatment of CCL and PL aims to re-establish joint kinematics, resolve pain, and return the patient to full function [7]. The majority of dogs with CCL injury and many with PL undergo surgery [5,7]. Evidence for nonsurgical CCL management is lacking, though the fundamental principles of rehabilitation and tissue healing should be applied through conservative management; however, the rehabilitation period is possibly prolonged relative to postoperative rehabilitation [7,8,9].

Neuromuscular control of limb motion is essential for normal walking to occur [9]. Shumway-Cook and Woollacott [10] state that optimized motor control is required for the exact modulation of the activation and coordination of muscles to produce functional, controlled movements. Gait abnormalities are common, including increased stifle flexion and reduced weight bearing due to pain or instability associated with stifle injury [5,11,12]. Joint degeneration affects the control of muscle forces, muscle firing patterns, balance, and proprioception hence the importance of re-establishing neuromuscular and postural control for returning the individual to normal stifle function, daily activity, and quality of life [5,13].

Balance, postural sway, and stability are components of postural control, referring to the ability to control the center of mass in relation to its base of support, i.e., controlling the position in space for orientation and stability [10]. Horak’s theoretical balance framework describes balance as a complex skill involving six interacting systems (Figure 1), all consisting of independent neurophysiological mechanisms controlling a specific aspect of postural control [14,15]. A recent study [16] focused on postural sway in dogs while standing, observing a larger movement in older dogs than younger dogs. Moreover, they found that aging was associated with muscle hypotrophy and reduced strength, and this could indicate that balance may be associated with muscle hypotrophy [16]. Carrillo et al. [17] demonstrated a higher sway in lame dogs with CCL rupture and elbow dysplasia compared to healthy ones; hence, muscle strength could be of importance for balance control in dogs with stifle injury. Recent studies [18,19,20,21] measured postural sway by observing changes in the center of pressure (COP), using force or pressure plates, in both the mediolateral and craniocaudal directions. Lutonsky et al. [19] used three-dimensional circular movements to challenge standing balance in healthy dogs and found a significant increase in multiple COP parameters with greater movement in the mediolateral direction than craniocaudal. Dogs with arthrosis of the stifle joint have shown an increase in mediolateral movements compared to healthy dogs [21], while dogs with osteoarthritis of the elbow or hip joint indicated compensatory changes in the COP within the paws [20].

Therapeutic exercise is aimed at facilitating recovery and assisting a more rapid return to function [22,23], promoting neuromusculoskeletal components of physical capacity [24]. Balance and weight-bearing exercises, such as three-legged standing, are used for a wide range of conditions (e.g., osteoarthritis, CCL rupture, fractures, and neurological conditions) and are considered important in canine rehabilitation [11,24,25,26,27]. Several studies have reported the benefits of postoperative rehabilitation after CCL surgery [7,25,28,29,30], though the specific effects of most exercises are unknown [13,31], and comparing and predicting the best outcome is challenging [29,30]. Many veterinary surgeons refer canine patients for rehabilitation, and canine rehabilitation professionals implement these therapeutic exercises in daily practice [32,33].

The rehabilitation process and therapeutic exercises rely on the principles of specificity and overload, and the dosage will determine the size of the effect [34,35,36]. In general, rehabilitation programs start with low-level activities progressing to a higher level [7,9]. All tissues respond differently to injury, inactivity, and remobilization. However controlled and progressive loading aids healing and challenges the body to build strength, though loading should be based on individual ability and activity without compensation and fatigue [7,31]. Rebuilding joint stability and restoring proprioception, while reducing pain is important in optimizing physical capacity [31,37] and in returning the dog to optimal activity levels as safely as possible [1,38].

The benefits of postoperative rehabilitation in the clinic, based on fundamental principles of rehabilitation, have been described in earlier studies [7,25,28,29,30]; however, what happens when we add a home exercise protocol of therapeutic exercise?

The aim of this study was to evaluate the effect of an add-on progressive therapeutic exercise protocol on balance control and function in dogs following stifle injury compared to a standard rehabilitation program. We hypothesized that progressive therapeutic exercise decreased the difference in static body weight distribution between hindlimbs, improved balance control, decreased pain-related disability, and improved stifle function.

## 2. Materials and Methods

### 2.1. Study Design

This study was a randomized controlled clinical trial consisting of two groups of consecutively recruited dogs with stifle injury.

### 2.2. Dogs

Thirty-three privately owned dogs, diagnosed and referred by a veterinarian to a veterinary physiotherapist at IVC Evidensia Animal Hospital Malmö, Sweden (November 2022 to June 2023), were enrolled in this study. Descriptive data of the dogs are illustrated in Table 1. The owners were informed about the study and the possibility of withdrawing their dog from participation at any time. The inclusion criterion was a veterinary-diagnosed stifle injury, while the exclusion criteria were other injuries affecting weight bearing on other limbs, infection, or FAS (fear, anxiety, and stress) level 3 or above [39]. The study was approved by the Animal Ethics Committee, Uppsala, Sweden (Dnr 5.8.18-17476/2022), and by the University of Liverpool Research Ethics Committee (Ref. 12167).

### 2.3. Procedure

The dogs were randomized to an intervention group or control group with the merged block randomization procedure [40]. They were randomized in groups of ten, using the software MERGEDBLOCKS version 1.1.0 (www.stephanievanderpas.nl/software, accessed on 17 October 2022). Both groups followed a standard rehabilitation protocol for stifle injury based on hydrotherapy, joint mobility, soft-tissue treatment, and home regime with activity restrictions and leash walking, performed and evaluated regularly by the veterinary physiotherapist (Appendix A). After measuring the baseline demographics and outcome data, the dogs started their protocols on their first visit with the veterinary physiotherapist. If surgery was conducted, the dogs started approximately 10–14 days following surgery; when being conservatively treated, the dogs started at their first visit after being diagnosed with stifle injury by the veterinary surgeon. Rehabilitation sessions in a clinical setting were part of the dogs’ veterinary treatment plan, in consultation with a team of animal health care professionals together with the dog owners. After 12 weeks, measurements of the outcome data were repeated.

The owners of the dogs in the intervention group were given a home exercise protocol of progressive therapeutic exercise to be performed two times daily (approximately 5–10 min per session) after instructions from the veterinary physiotherapist, as well as a written copy of the training protocol. The protocol included four different levels based on three-legged standing, developed to address the six systems of balance control (Figure 1): (I) biomechanical constraints (four-legged to three-legged and two-legged stances, repetitive weight shifting), (II) anticipatory postural adjustments (four-legged to three-legged and two-legged stances, repetitive weight shifting), (III) postural responses (weight shifting, three-legged stance with push), (IV) sensory orientation (standing on unstable surface, incline), (V) stability limits (weight shifting, three-legged stance with push), and (VI) stability in gaits (addressed using a standard rehabilitation protocol with hydrotherapy and controlled leash walking) [14,15]. The exercises progressed from less challenging to more challenging according to the individual functional level at assessment, as determined by the veterinary physiotherapist. The home exercise protocol is described in Appendix B.

During the study, all physiotherapeutic assessments, rehabilitation sessions, and data collection activities were conducted by the same veterinary physiotherapist (I.B.). During the study period, there were regular clinical re-assessments and rehabilitation sessions.

#### 2.3.1. Demographic Data

Weight, body condition score (BSC) [41], and muscle condition score (MCS) [42,43] were recorded for all dogs at baseline. The body condition score is a nine-point scale system used to assess body composition, where an ideal weight corresponds to a score of four or five, with lower scores being classified as “too thin” and higher scores as “overweight” or “obese” [41,44]. The muscle condition score is graded from 0 to 3, where low scores mean “normal muscle mass”, and higher scores correspond to “mild, moderate, or severe muscle loss” [42,43]

#### 2.3.2. Outcome Measures

Difference in body weight distribution between the hindlimbs was measured in static standing at a Companion Stance Analyzer (PetSafe Stance Analyzer, version 1.20.0.0, 2012, Companion Animal Health, New Castle, DE, USA). The dog was in a squared stance with its head and eyes forward towards the owner. Ten recordings (one recording/second) were registered during four-legged standing, calculating the mean proportion of total body weight for each hindlimb. The absolute difference between hindlimbs was used for the analysis.

Balance control was measured in static standing as the difference in mediolateral sway between the hindlimbs while in a three-legged stance (measuring movement in cm during 4 s of continuous three-legged standing). Having the dog standing as balanced as possible with head and eyes focused forward at the handler, each hindlimb lifted in position for 5 s. Video was recorded (using GoPro camera HERO4, GoPro Inc., San Mateo, CA, USA) from a caudocranial view at the height of the tuber ischii, exposing the frontal plane. Mediolateral movements were analyzed using the software Kinovea 0.9.5 (https://www.kinovea.org/, accessed on 22 March 2023) as a valid and reliable tool to measure distances up to 5 m from an object [45,46] (Figure 2a,b). Markers (soft, white, and sticky furniture pads with a diameter of 18 mm marked at the center) were placed at L7 and caput metatarsale V bilaterally, representing landmarks used for measurements [47]. The dogs were video recorded, while measuring static body weight distribution, standing at the Companion Stance Analyzer.

The Canine brief pain inventory (CBPI) was used to assess the interference of pain with the dogs’ function. The CBPI detects changes in the impact of pain on function in dogs with osteoarthritis and is a validated and quantifiable measure of the owners’ perceptions [48]. The CBPI contains 10 questions with a rating from 0 to 10, with four questions concerning the severity of pain and six questions regarding how the pain interferes with daily activities. A mean value providing the pain interference score (0 = pain does not interfere, 10 = pain completely interferes) was used in this study, i.e., low scores indicate a low impact of the pain [48]. Additionally, there is a validated Swedish-translated version appropriate for use in this study [49].

The Finnish canine stifle index (FCSI) was used to measure stifle functionality as a testing battery with eight items divided into active (compensations in positions of sitting and lying and symmetry of thrust of hindlimbs getting up from positions of sitting and lying) and passive (assessment of muscle symmetry, measurement of symmetry in static weight bearing between hindlimbs using bathroom scales, and measured flexion and extension of the stifle joint using a goniometer) components used to quantify the level of the dog’s stifle function [50]. The FCSI generates a numerical index ranging from 0 to 263, with cut-off values at 60 (separating the “adequate” and “compromised” functions) and 120 (distinguishing “severely compromised” from “compromised” and “adequate”), which has been tested and found to be reliable for dogs with stifle injury [51].

### 2.4. Statistical Analysis

Descriptive data are presented as the mean, with the range and standard deviation (SD) describing the spread of the data, i.e., body weight and age, as well as the median with range describing the spread of the data, i.e., BCS and MCS. Data on the outcomes were tested for normal distribution using the Shapiro–Wilk test. Comparisons of the outcome measures between the intervention and control groups were conducted using the Mann–Whitney U-test. The paired equivalent Wilcoxon signed-rank test was used to assess the same outcome measures for improvement over time. All statistical analysis was performed using IBM SPSS Statistics (version 28.0.1.1). The significance level for all tests was set to *p* ≤ 0.05. Effect size estimates were calculated to evaluate the magnitude of the effect of the intervention, with the correlation coefficient (*r*) being a value ranging from −1.00 to 1.00 [52,53]. The intervals of *r* = 0.1–0.3 (small effect), *r* = 0.3–0.5 (intermediate effect), and *r* > 0.5 (strong effect) are according to Cohen [54]. The correlation coefficient was calculated using the following formula [52,53]:$$r=\frac{Z}{\sqrt{N}}$$

The *Z*-scores were estimated with the Mann–Whitney U test and Wilcoxon signed-rank test, respectively. *N* is the total number of observations.

## 3. Results

### 3.1. Intervention and Control Groups

Of the 33 dogs entering the study, 26 dogs (14 male, 12 female) finished the study (Figure 3). The intervention group included 14 dogs (9 male, 5 female), and the control group included 12 dogs (6 male, 6 female). Descriptive data on the intervention and control groups are illustrated in Table 1. There were no significant differences in the basic characteristics between those that were lost/excluded during the study and those that remained.

### 3.2. Effect of a Therapeutic Exercise Protocol within and between Intervention and Control Groups

For all outcome measures, there were no significant differences between the groups at baseline. After the intervention period of 12 weeks, the median difference in static body weight distribution between the hindlimbs was significantly lower (*p* = 0.046), with an intermediate effect in favor of the intervention group compared with the control group (median = 2.5% vs. 5.5%; Figure 4a). Moreover, the median CBPI pain interference score was significantly lower (*p* = 0.004), with a strong effect in the intervention group compared with the control group (median = 0.0 vs. 0.9; Figure 4c), and the median score of the FCSI with intermediate effects (*p* = 0.02) was in favor of the intervention group compared with the control group (median = 25 vs. 75; Figure 4d). However, there was no significant difference (*p* = 0.572) for balance control (i.e., mediolateral sway) between the intervention group and the control group after 12 weeks (Figure 4b). The statistics are presented in Table 2. Both the intervention group and the control group improved in all outcome measures with intermediate to strong effects, except for balance control (i.e., mediolateral sway). The statistics are presented in Table 3.

## 4. Discussion

This study is a randomized controlled trial and, to our knowledge, one of the few experimental studies to evaluate the effect of therapeutic exercise, as an add-on intervention, on balance, function, and static body weight distribution. We hypothesized that progressive therapeutic exercise decreased the difference in static body weight distribution between the hindlimbs, improved balance control, decreased pain-related disability, and improved stifle function. The results revealed that both the intervention and control groups showed a statistically significant improvement during the 12 weeks of rehabilitation regarding static body weight distribution between the hindlimbs, pain-related functional disability, and stifle function. There was no statistically significant effect on balance control (i.e., mediolateral sway) either within the groups or between the groups, with a small effect size estimate; however, we must consider the rather small sample group and the method of measurement not being tested or validated.

Previous studies [25,28,29,30] have shown the significance of early rehabilitation after stifle injury. The results of this study were in congruence with these, as well as with Kirkby Shaw et al. [7], and their proposed general guidelines for post-CCL surgery rehabilitation based on tissue healing, individual assessments, clinical reasoning, and functional goals, with therapeutic exercise being the foundation of rehabilitation. However, in contrast, this study evaluated the more specific effects of therapeutic exercise as a home exercise protocol.

The training principles of specificity, overload, and progression are essential for rehabilitation and therapeutic exercises. Hence, this protocol of therapeutic exercises should not be generalized and used without continuously assessing physical capacity and pain and progressing the exercises accordingly to challenge tissues. However, it must be carefully adjusted to tissue healing, strength, and functional ability. Kirkby-Shaw et al. [7] introduced four fundamental principles of rehabilitation: (I) tissues follow a certain pattern of healing (acute/inflammatory phase, subacute/reparative phase, and chronic/remodeling phase), (II) individualized treatment plans adjusted frequently based on an assessment of tissue healing, strength, and functional abilities/limitations, (III) specific, measurable, attainable, and relevant goals that consider not only the injury but the whole animal, which should be modified throughout the phases of healing, and (IV) the foundations of physical rehabilitation: pain management, therapeutic exercise, manual therapy, and guided return to activity. The veterinary physiotherapist has an essential role in applying rehabilitation principles and in the progression of therapeutic exercise. For example, hydrotherapy can be used in different stages of rehabilitation [55,56,57]. The early introduction of underwater treadmill exercise (subacute/reparative phase), with slow speed and adjusting the depth of water according to the preferred ROM, promotes limb loading and activation of hypotrophied muscles while also improving proprioception and balance [55,56]. The home exercise protocol described in Appendix B, as mentioned earlier, was developed to address the six systems of balance control (Figure 1); as described by Witter and Bockstahler [58] and Millis and Levine [59], they are used to promote limb loading, activate muscles, and reduce pain by promoting neuromuscular interactions. While following the four principles of rehabilitation, adapted to the patient’s current status, we believe the exercises are safe for early introduction. Since the FCSI is reliable for measuring stifle functionality, this functional level of assessment could, perhaps, be used to standardize the progression of exercise.

### 4.1. Clinical Implications

The intervention group showed a significantly greater improvement than the control group regarding static body weight distribution between the hindlimbs, pain-related functional disability, and stifle function. The FCSI cut-off value of 60 was used to separate “adequate” from “compromised” stifle function [51], and it is of important clinical relevance in this study that all dogs in the intervention group within the IQR are considered to have adequate stifle function (median = 25, IQR = 9.4–40.6) after 12 weeks of rehabilitation, compared to the compromised stifle function of the control group (median = 75, IQR = 31.3–87.5). To our knowledge, this is the first study to use the FCSI as an outcome measure for stifle function investigating the effect of a physiotherapeutic rehabilitation protocol. Considering static body weight distribution between hindlimbs, Hyytiäinen et al. [60] found it to be a reliable and objective method of measurement in dogs with hindlimb osteoarthritis and established the normal difference between the hindlimbs of healthy dogs to be 3.3% (SD = 2.7%). After the intervention, the intervention group in this study reached a mean difference of 2.5%, well below the reference of 3.3%, while the control group reached a difference of 5.5%, which is close to the cut-off value of 6% (3.3%, SD = 2.7%). We do not have comparable cut-off values for the CBPI pain interference score, but from our effect size estimate, we have an intermediate to strong clinical effect for all of the outcome measures above, increasing the magnitude of our results.

There was no statistically significant effect on balance control (i.e., mediolateral sway) either within the groups or between the groups. Hence, this method needs to be evaluated thoroughly in future studies, considering the eventual learning effect of lifting limbs in a specific order during measurements. We measured sway in the mediolateral direction while the craniocaudal shift was not assessed. We did not control for how much weight support the dog received from the hand of the physiotherapist, which might have affected the results. However, since the dogs were randomized into groups and handled similarly, it is not likely that there is a difference between the groups, and our conclusions regarding the treatment effects are still valid. Performing a qualitative movement assessment using the video recordings, it was evident that the dogs used different strategies to handle pain, reducing the weight-bearing capacity and strength of the affected limb, which should be further evaluated. If we can evolve our ability to quantify balance control, it will possibly improve our capability to treat the different mechanisms involved. Standardized settings and validated measurement methods of balance control using a standard video camera or smartphone, and analyzing data using software like Kinovea version 0.9.5 (valid and reliable for measuring distances and angles [45,46]), could be an affordable and applicable method to use in a clinical setting.

### 4.2. Methodological Considerations and Future Studies

To monitor rehabilitation interventions in dogs with stifle injury, validated and clinically relevant outcome measurements are of great importance. In this study, most of the outcome measures used are considered objective, reliable, and valid for evaluating function, balance, and muscle strength. Static body weight distribution between the hindlimbs, the CBPI, and the FCSI are all methods/measurements tested and found to be reliable for evaluating dogs with stifle injury [48,50,60,61,62,63,64,65,66], and Hyytiäinen et al. [51] indicate that the FCSI is more sensitive to stifle dysfunction over other dysfunctions, increasing the internal validity of this study.

The FCSI was developed as a testing battery to evaluate the level of stifle function and was found to be reliable, though has not yet been used as an outcome measure in a clinical study [50]. The CBPI is an owner-assessment questionnaire developed to quantify the owner-assessed severity of pain and pain interference on daily activities [48], and it has been used to evaluate the effect of analgesic therapy, medical treatment, and acupuncture in dogs with pain associated with osteoarthritis [48,61,67]. Digital scales have been used to measure the static weight distribution of the hindlimbs and the difference between the hindlimbs (expressed in percentage of total body weight) [60,63,64,65,66]. Hyytiäinen et al. [60] found it to be a reliable and objective method of measurement in dogs with hindlimb osteoarthritis. A study by Wilson et al. [62] found that the stance analyzer allowed for repeatable measurements of the weight distribution of the hindlimbs of dogs with hindlimb lameness. Most studies, including the present study, are focused on the compensation strategies of the hindlimbs only. The role of the forelimbs in balance remains unclear and is something we would like to study simultaneously, but methodological studies are needed in this field.

The effects of other co-interventions, such as the use of pain medication during the rehabilitation period, are unclear, but due to the effectful randomization, it is not plausible that these co-interventions could have influenced the between-group effects.

Earlier studies [7,25,28,29,30] have shown the significance of rehabilitation after stifle injury, but this study also shows the importance of performing home exercise between the sessions with the veterinary physiotherapist, and that it does not require more than 5–10 min two times daily.

Most dogs participating in this study had surgery, and only one dog was managed conservatively, which can be considered a limitation. Even though the evidence for nonsurgical CCL management is lacking and the rehabilitation period may be prolonged [7,8,9], fundamental principles of rehabilitation were followed and there were no substantial differences in the results after 12 weeks of rehabilitation for the dog going through nonsurgical management.

An important limitation of this study was that the same veterinary physiotherapist conducted all physiotherapeutic assessments, rehabilitation sessions, and data collection activities, as well as the fact that they were not blinded to which group the patients were allocated. This could have affected the data management’s validity. Furthermore, the owners were also not blinded to which group they were allocated. By clinical experience a common limitation is owner compliance and whether the exercises were performed with good form, i.e., understanding when a dog starts compensating because of fatigue. These issues were addressed by taking time to show the exercises to owners and having them repeat exercises with their dogs in rehabilitation sessions before sending them home and repeating them at their next session.

Another limitation of this study was the rather small sample size. Further, there are external factors that could have affected the measurements of static body weight distribution between the hindlimbs, as well as control of balance measuring mediolateral movement. The position of the owner, the side of the leash, and, possibly, the placement of the stance analyzer (e.g., close to a wall) have been shown to impact the results of weight distribution [68,69]. Small shifts in movement by the dogs, such as weight shifting or head turns, are impossible to control and may contribute to errors in measurements. The time frame of 12 weeks could be seen as too short a rehabilitation time, since not all dogs regained full function. However, it is for practical reasons not possible to have longer rehabilitation programs. In general, dogs are possibly not fully rehabilitated within 12 weeks of stifle injury. This study evaluated short-term results and the long-term effects remain unknown. Previous studies showed that there is a high rate of CCL rupture to the contralateral limb, with a risk ranging from 22 to 54% after 10 to 17 months from the first diagnosis [70,71,72], and Muir et al. [73] determined a median time to contralateral injury of 947 days. Hence, for future studies, there is a need to study the rehabilitation of dogs with stifle injury over a longer period, with a follow-up within six months to a year.

Concerning the external validity, the results of this study could be seen as a representative sample of a larger population in Sweden and, possibly, even for other comparable countries. This study investigated the effect of therapeutic exercise in combination with an in-clinic standard rehabilitation protocol for stifle injury. We do not know the efficiency of the home exercise protocol alone, if one does not have the ability to regularly visit an animal physiotherapist for evaluation, assessments, and in-clinic rehabilitation.

In future studies, it would be interesting to further investigate the muscle activation and strength of the quadriceps, gastrocnemius, and biceps femoris, which are all muscles that require coordination to provide passive and dynamic stability [4], while reduced timing may increase the susceptibility to injury [5].

## 5. Conclusions

This study indicates that a progressive therapeutic exercise protocol based on three-legged standing and targeting balance and postural- and neuromuscular control decreased the difference in static body weight distribution between the hindlimbs, decreased pain-related functional disability, and improved stifle function in dogs diagnosed with stifle injury after 12 weeks of daily home exercise. The effects were statistically significant and showed intermediate to strong effects. Since therapeutic exercise is already widely implemented in daily practice as an important part of rehabilitation, the clinical relevance of our results verifies the use of these exercises. Further, this study indicates that there was no difference and a small effect on the balance control in the mediolateral direction of the therapeutic exercise protocol. The rather small sample group and unvalidated measurement method must be considered and may explain the results; hence, we suggest balance control in dogs with stifle injuries to be tested more thoroughly in future studies. The results are highly clinically relevant, since all dogs in the intervention group reached an adequate level of stifle function following daily therapeutic exercise for three months.

## Figures and Tables

**Figure 1 animals-14-00092-f001:**
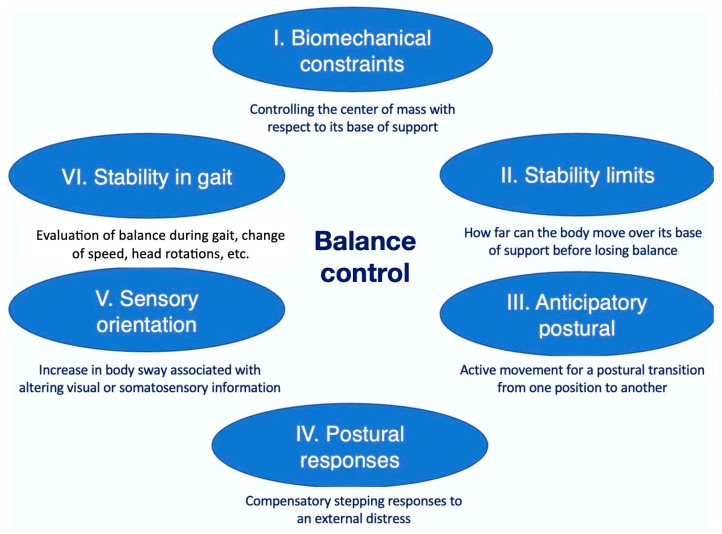
Horak’s theoretical framework, showing six interacting components of balance control. Figure adapted with permission from Horak [14] and Horak et al. [15].

**Figure 2 animals-14-00092-f002:**
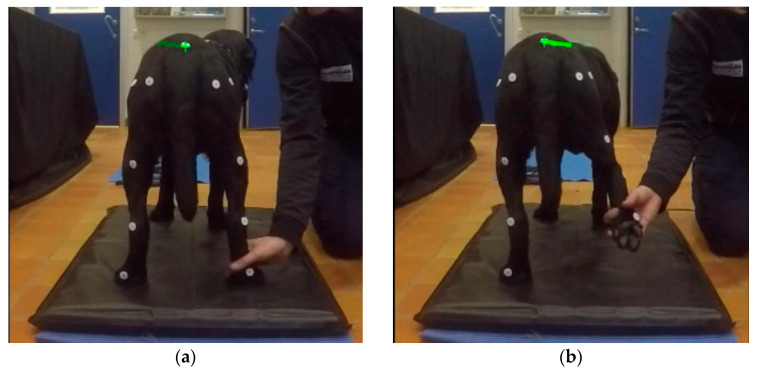
Balance control. The video recording and measurements were performed while in (**a**) the baseline position with a four-legged stance just before lifting the limb and in (**b**) a three-legged standing position with the hindlimb lifted showing the mediolateral movement from the baseline position. The green line indicates the displacement of L7 while the right hindlimb is lifted.

**Figure 3 animals-14-00092-f003:**
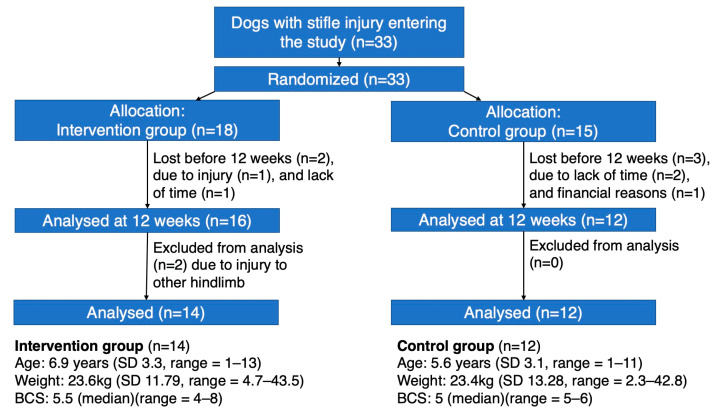
Flowchart of the participating dogs in the study, showing the number of participants randomly assigned to the intervention and control groups, the number of participants lost/excluded before 12 weeks, and the number of participants excluded from the analysis, as well as the basic characteristics of the sample intervention and control groups.

**Figure 4 animals-14-00092-f004:**
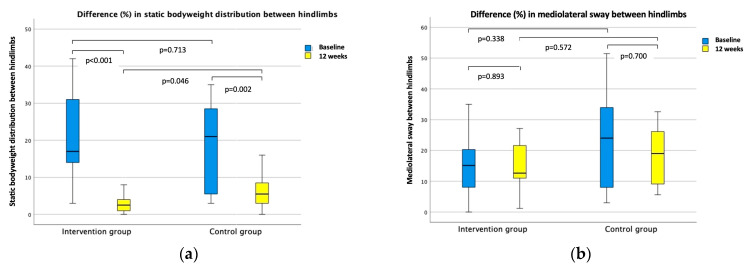
Primary and secondary outcomes. Boxplots present the median difference within the intervention and control groups, as well as the median difference between the intervention and control groups: (**a**) difference in static body weight distribution between the hindlimbs; (**b**) difference in mediolateral sway between the hindlimbs; (**c**) canine brief pain inventory (CBPI) pain interference score; (**d**) Finnish canine stifle index (FCSI).

**Table 1 animals-14-00092-t001:** Descriptive data of the intervention and control groups.

**Intervention Group**								
**Sex**	**Age**	**Breed**	**Weight**	**BCS**	**MCS**	**Type of Injury**	**Intervention**	**Medication**
F	5	Labrador Retriever	31.6	7	1	CCL + meniscus	TPLO	Paracetamol
F	1	Miniature poodle	4.7	5	1	MPL	TBR + TTT	NSAID
F	10	Jack Russel Terrier	6.5	6	1	CCL + meniscus	TPLO	NSAID
M	13	Mixed breed	26.1	6	1	CCL	TPLO	NSAID
M	8	American Staffordshire Terrier	32.3	6	0	CCL	TPLO	NSAID
M	11	Bichon Havanais	6.7	5	0	CCL	Conservative	NSAID
M	9	American Staffordshire Terrier	30.8	6	0	CCL + meniscus	TPLO	NSAID
F	5	Staffordshire Bullterrier	17.4	6	0	MPL	TBR + TTT	NSAID
F	8	Mixed breed	21	5	0	CCL + meniscus	TPLO	NSAID
M	7	Golden Retriever	43.5	8	2	CCL	TPLO	NSAID
M	5	American Staffordshire Terrier	30	4	0	CCL	TTA	NSAID
M	7	French Bulldog	17.1	7	0	CCL	TTA	NSAID
M	5	Mixed breed	36	5	0	CCL	TTA	NSAID
F	1	Mixed breed	19.4	5	0	CCL	TPLO	NSAID
M	2	Staffordshire Bullterrier	17.8	5	0	MPL	TTT	NSAID
F	5	Mixed breed	25.2	5	0	CCL	TTA rapid	NSAID
M	6	Mixed breed	35.8	5	1	CCL	TPLO	NSAID
M	10	Jack Russel Terrier	9.6	7	0	CCL	TPLO	NSAID
	6.6 (3.25) (1–13)		22.9 (6.16) (4.7–43.5)	5.5 (4–8)	0 (0–2)	15 CCL (3 with meniscus) + 3 MPL		
**Control Group**								
**Sex**	**Age**	**Breed**	**Weight**	**BCS**	**MCS**	**Type of Injury**	**Intervention**	**Medication**
M	9	Mixed breed	34	6	1	CCL	TPLO	NSAID
F	4	Cavalier King Charles Spaniel	9.2	6	1	MPL	TTT	NSAID
F	9	Mixed breed	31	6	1	CCL + meniscus	TPLO	NSAID
M	7	Beagle	19.2	6	0	CCL	TPLO	NSAID
M	5	Mixed breed	42.8	5	1	CCL	TPLO	NSAID
M	5	Pomeranian	3.8	5	2	CCL + MPL	TPLO	NSAID
F	3	Dogo Argentino	40	5	1	CCL	TPLO	NSAID
F	11	Mixed breed	21	6	1	CCL + meniscus	TPLO	NSAID
F	1	American Staffordshire Terrier	26	5	0	CCL	TTA	NSAID
F	2	Pomeranian	2.7	5	1	MPL	TWR + TTT	NSAID
F	4	Mixed breed	31	5	0	CCL + meniscus	TPLO	NSAID
M	7	Cardigan Welsh corgi	19.5	5	1	CCL + meniscus	TPLO	NSAID
F	9	Mixed breed	30.2	5	0	CCL	TPLO	NSAID
M	1	Standard poodle	8	5	0	MPL	TTT	NSAID
M	9	Australian Shepherd	22.2	5	1	CCL	TPLO	NSAID
	5.7 (3.13) (1–11)		22.7 (9.62) (2.3–42.8)	5 (5–6)	1 (0–2)	11 CCL (4 with meniscus) + 3 MPL + 1 CCL + MPL		

Continuous variables (age and weight) are presented with the mean (SD) (range) and categorical variables (BCS and MCS) with the median (range). F = female, M = male, BCS = body condition score, MCS = muscle condition score, CCL = cranial cruciate ligament, MPL = medial patellar luxation, TPLO = tibial plateau leveling osteotomy, TTA = tibial tuberosity advancement, TBR = trochlear block recession, TWR = trochlear wedge recession, TTT = tibial tuberosity transposition, SD = standard deviation, and NSAID = nonsteroidal anti-inflammatory drug.

**Table 2 animals-14-00092-t002:** Between-group effects.

			Intervention		Control			
Outcome Measure	Timepoint	*n*	Median (IQR)	*n*	Median (IQR)	*p*-Value	z-Score	Effect Size (r)
Static BWD difference between hindlimbs	Baseline	14	17% (13.3–31.8)	12	21% (5.3–28.8)	0.713	−0.386	−0.076
	12 weeks	14	2.5% (1.0–4.5)	12	5.5% (3.0–8.8)	0.046	−1.994	−0.391
Balance control (mediolateral sway)	Baseline	13	15.1% (5.9–22.6)	11	24.0% (8.0–35.9)	0.338	−0.985	−0.201
	12 weeks	14	12.6% (4.9–23.0)	11	19.0% (9.1–27.0)	0.572	−0.602	−0.120
CBPI pain interference	Baseline	14	3.6 (1.8–4.9)	12	4.9 (2.8–6.5)	0.197	−1.313	−0.258
	12 weeks	14	0.0 (0.0–0.2)	12	0.9 (0.1–1.8)	0.004	−2.806	−0.550
FCSI	Baseline	14	187.5 (125–190.7)	12	193.7 (150–209.4)	0.220	−1.252	−0.246
	12 weeks	14	25.0 (9.4–40.6)	12	75.0 (31.3–87.5)	0.020	−2.303	−0.452

BWD = body weight distribution, CBPI = canine brief pain inventory, FCSI = Finnish canine stifle index, and IQR = interquartile range.

**Table 3 animals-14-00092-t003:** Within-group effects.

Outcome Measure	Group	Timepoint	*n*	Median (IQR)	*p*-Value	z-Score	Effect Size (r)
Static BWD difference between hindlimbs	Intervention	Baseline	14	17.0% (13.3–31.8)	<0.001	−3.306	−0.624
		12 weeks	14	2.5% (1.0–4.5)			
	Control	Baseline	12	21% (5.3–28.8)	0.002	−2.805	−0.527
		12 weeks	12	5.5% (3.0–8.8)			
Balance control (mediolateral sway)	Intervention	Baseline	13	15.1% (5.9–22.6)	0.893	0.175	−0.034
		12 weeks	14	12.6% (4.9–23.0)			
	Control	Baseline	11	24.0% (8.0–35.9)	0.700	−0.445	−0.095
		12 weeks	11	19.0% (9.1–27.0)			
CBPI pain interference	Intervention	Baseline	14	3.6 (1.8–4.9)	<0.001	−3.297	−0.623
		12 weeks	14	0 (0–0.2)			
	Control	Baseline	12	4.9 (2.8–6.5)	<0.001	−2.936	−0.599
		12 weeks	12	0.9 (0.1–1.8)			
FCSI	Intervention	Baseline	14	187.5 (125–190.7)	<0.001	−3.3	−0.624
		12 weeks	14	25.0 (9.4–40.6)			
	Control	Baseline	12	193.7 (150.0–209.4)	<0.001	−3.063	−0.625
		12 weeks	12	75.0 (31.3–87.5)			

BWD = body weight distribution, CBPI = canine brief pain inventory, FCSI = Finnish canine stifle index, and IQR = interquartile range.

## Data Availability

The data presented in this study are available on request from the corresponding author. The data are not publicly available due to ethical and privacy limitations based on the consent provided by participants.

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
