# Peer review of "The Effect of Therapeutic Exercise on Body Weight Distribution, Balance, and Stifle Function in Dogs following Stifle Injury"

_animals, 2023, doi:10.3390/ani14010092_

Round 1

Reviewer 1 Report

Comments and Suggestions for Authors

Dear authors

Thank you for forwarding this interesting and important article.

I have the following comments:

Introduction: This is well written and interesting, however I think that some publications are missing regarding PS (Lutonsky C, 2023, Charalambous D 2023, Virag Y 2022, Bieber B 2022, Charalambous D 2022, Reicher B 2022) - they could be interesting especially for the CCLR topic.

Procedere: You write that the protocol was started on the first visit. This is a little unclear to me. Do you also mean appointments that took place before the surgery? So, for example, the dog came to you, then started the protocol and was then surgically treated? Because all but one dog was surgically treated.

Outcomes

I think it's a pity that you only used data from the hind legs. We know that compensation can also occur in the front legs and I think that important information has been lost here.

I'm generally not a fan of the stance measurements, even tiny movements of the head or moving the leg slightly outwards/inside changes the load massively. How did you address the problem?

Could you please specify the exact times of the measurements? When was the baseline measurement taken, when was the control measurement taken? I understand from the results that only a control measurement was taken after 12 weeks?

Regarding the lack of effect on balance: do you think that the evaluation of the mediolateral sway was sufficient? As I have already written, there will probably have been compensatory effects, which could also be expressed in the craniocaudal sway, for example. The evaluation of a single parameter is probably not sufficient. It might also have been advantageous to determine the body CoP. Furthermore, although it is very difficult to measure, lifting one leg may not be the best way to unbalance the dog. There may be animals that put more weight in your hand and some that don't like it when your paw is touched (just as examples). I think there are better ways to challenge balance. This should be discussed.

Author Response

Authors response to reviewer 1

Introduction: This is well written and interesting, however I think that some publications are missing regarding PS (Lutonsky C, 2023, Charalambous D 2023, Virag Y 2022, Bieber B 2022, Charalambous D 2022, Reicher B 2022) - they could be interesting especially for the CCLR topic.

Thank you for highlighting these studies, we have now added the following in the Introduction on page 2, line 87-94.

Recent studies (18–21) have measured postural sway by observing changes in the center of pressure (COP), using force or pressure plates, in both mediolateral and craniocaudal direction. Lutonsky et. al. (19) used three-dimensional circular movements to challenge standing balance in healthy dogs and found a significant increase in multiple COP parameters with greater movement in the mediolateral direction than craniocaudal. Dogs with arthrosis of the stifle joint have shown an increase in mediolateral movements compared to healthy dogs (21), while dogs with osteoarthritis of the elbow or hip joint indicated compensatory changes in the COP within the paws (20).

Procedere: You write that the protocol was started on the first visit. This is a little unclear to me. Do you also mean appointments that took place before the surgery? So, for example, the dog came to you, then started the protocol and was then surgically treated? Because all but one dog was surgically treated.

Thank you for pointing this out, we have rephrased the sentence describing this in the first paragraph under “2.3 Procedure”, page 7, line 155-160.

After measuring baseline demographics and outcome data, the dogs started their protocols on their first visit with the veterinary physiotherapist. If surgery was conducted, the dogs started approximately 10-14 days following surgery, when being conservatively treated, the dogs started at their first visit after being diagnosed with stifle injury by the veterinary surgeon.

Outcomes

I think it's a pity that you only used data from the hind legs. We know that compensation can also occur in the front legs and I think that important information has been lost here.

The authors agree that compensation also occurs in the forelimbs, and important information can be missed not involving this information in this study. However, to our knowledge, there are no comparable studies in which the weight distribution of forelimbs in relation to hindlimbs is evaluated in dogs with stifle injury. Hence, we focused on the study of hindlimbs where Hyytiäinen et. al. (60) provided a reliable and objective method of measurement, stating a normal difference between the hindlimbs of healthy dogs, allowing us to compare our measurements of static body weight distribution between hindlimbs with previous studies. Moreover, Hyytiäinen et. al. (50,51) have used body weight bearing of the hindlimbs as one of eight items in FCSI and showed that FCSI is reliable for dogs with stifle injury.

Based on your comment, we have added this sentence to the discussion, page 14, line 393-395

“Most studies, including this present study, are studying compensation strategies of the hindlimbs only. The role of the forelimbs in balance remains unclear and something we would like to study simultaneously, but methodological studies are needed in this field.”

I'm generally not a fan of the stance measurements, even tiny movements of the head or moving the leg slightly outwards/inside changes the load massively. How did you address the problem?

The authors are aware of this problem while measuring weight distribution. Phelps et al (2007) studied the effect of environment, location, and confinement on measurements of static load distribution. They found the position with the dog placed in the center of the room with the dog’s head held neutrally was the most consistent. Also, measurements of each position showed no significant difference between days or sessions, suggesting that using a standardized placement of the dog, is a reliable method of measuring load distribution. Hence, we took our time to adjust the position of the dog standing on the stance analyzer making sure the dog was in a squared stance with its head and eyes forward towards the owner positioned in front of the dog. The stance analyzer was always placed in the middle of the room with no confinement on either side. We made repeated measurements of ten recordings, one recording/second, calculating the mean, while excluding outliers. Also, we know from earlier methodological studies on dogs with stifle injuries by Hyytiäinen et. al. (60), that the measurement error is around 3.3% (SD 2.7%) when measuring static weight bearing between hindlimbs. We believe this measurement error does not differ between the intervention and the control groups.

Could you please specify the exact times of the measurements? When was the baseline measurement taken, when was the control measurement taken? I understand from the results that only a control measurement was taken after 12 weeks?

Thank you for pointing this out, we can see this is unclear in our description of the procedure. We have rephrased the first paragraph “2.3 Procedure” on page 7, line 150-162, as follows:

“The dogs were randomized to an intervention group or control group, by the procedure of merged block randomization (40). They were randomized in groups of ten, using software MERGEDBLOCKS (www.stephanievanderpas.nl/software). Both groups followed a standard rehabilitation protocol for stifle injury based on hydrotherapy, joint mobility, soft-tissue treatment, and home regime with activity restrictions and leash walking, performed and evaluated regularly by the veterinary physiotherapist (Appendix A). After measuring baseline demographics and outcome data, the dogs started their protocols on their first visit with the veterinary physiotherapist. If surgery was conducted, the dogs started approximately 10-14 days following surgery, when being conservatively treated, the dogs started at their first visit after being diagnosed with stifle injury by the veterinary surgeon. Rehabilitation sessions were part of the dog’s veterinary treatment plan, in consultation with a team of animal health care professionals together with the dog owners, in a clinical setting. After 12 weeks, measurements of outcome data were repeated.” 

Regarding the lack of effect on balance: do you think that the evaluation of the mediolateral sway was sufficient? As I have already written, there will probably have been compensatory effects, which could also be expressed in the craniocaudal sway, for example. The evaluation of a single parameter is probably not sufficient. It might also have been advantageous to determine the body CoP. Furthermore, although it is very difficult to measure, lifting one leg may not be the best way to unbalance the dog. There may be animals that put more weight in your hand and some that don't like it when your paw is touched (just as examples). I think there are better ways to challenge balance. This should be discussed.

Regarding the newly developed balance measures, we have added and rephrased the last paragraph under ”4.1. Clinical implications”, page 14, line 357-373, in the discussion as follows:

“There was no statistically significant effect on balance control, i.e., mediolateral sway, neither within groups nor between groups. Hence the method needs to be evaluated thoroughly in future studies, considering the eventual learning effect of lifting limbs in a specific order during measurements. We measured sway in the mediolateral direction while the craniocaudal shift was not assessed. We did not control for how much weight support the dog got from the hand of the physiotherapist, which might have affected the results. However, since the dogs were randomized into groups and handled similarly, it is not likely there is a difference between the groups, and that our conclusions on treatment effects are still valid. Performing a qualitative movement assessment using video recordings, it was evident that dogs used different strategies to handle pain, reduced weight-bearing, and strength of the affected limb, which suggestively should be further evaluated. If we can evolve our ability to quantify balance control, it will possibly improve our capability to treat the different mechanisms involved. Standardized settings and validated measurement methods of balance control using a standard video camera or smartphone, and analyzing data in software like Kinovea (valid and reliable for measuring distances and angles (45,46), could be an affordable and applicable method to use in a clinical setting.”

We would also like to try and improve the measure of balance in future studies. We believe there are challenges in measuring balance while keeping it clinical, with difficulties in making procedures fully experimental while still being repeatable. Hence, this balance measure is a method that can be improved.

Reviewer 2 Report

Comments and Suggestions for Authors

General comments:

This is a very interesting study, well structured, with a notable hard work and effort! There are some issues and concerns that should be adressed, making this a publishable manuscript and with the possibility for further studies in the future.

Specific comments:

Line 121: How did the authors did the randomization process. Please state here.

Line 125: These dogs were diagnosed with what? Ligament rupture, patella luxation?? I think that in population presentation the authors should consider to caracterize according to etiology to reduce bias possibility. Also authors should refere if all dogs are from pos-surgery or only conservative treatment.

Line 142 - At admission what was the lameness degree of these patients? The authors should add this information in table 1.

Line 154: The authors should consider to add photos when describing the protocol.

Line 372: If only one dog was in conservative treatment, the authors should consider to exclude this case. 

In the discussion section the authors should discuss the early introduction of hydrotherapy and other possible exercises. Compare to other studies and bibliographic references.

Is this a study in a clinical setting? If so, the authors should state this.

Author Response

Authors response to reviewer 2

Specific comments:

Line 121: How did the authors did the randomization process. Please state here.

Randomization was made through merged block randomization as described in section “2.3. Procedure”, page 7, line 151-152. To clarify the method, we have added the sentence as follows:

“They were randomized in groups of ten, using software MERGEDBLOCKS (www.stephanievanderpas.nl/software).”

Line 125: These dogs were diagnosed with what? Ligament rupture, patella luxation?? I think that in population presentation the authors should consider to caracterize according to etiology to reduce bias possibility. Also authors should refere if all dogs are from pos-surgery or only conservative treatment.

We thank you for your input about characterizing according to etiology. All dogs in this study, according to inclusion criteria, were diagnosed with stifle injury, as described in Table 1. Our chosen outcome measures of static body weight distribution and FCSI are valid for measuring stifle injury and are not specific to different types of etiology. Also, CBPI is considered generic in dogs with joint diseases. We wanted to study the effect of therapeutic exercise on various stifle injuries, hence, we have chosen not to categorize the injuries further. There is a rather even spread in the intervention and control group, and the dogs are randomized to control for selection bias.

Line 142 - At admission what was the lameness degree of these patients? The authors should add this information in table 1.

Subjective lameness rating was not assessed at admission. Instead, we chose to use objective validated and reliable outcome measures for assessing pain and function by using CBPI, static weight bearing, and FCSI.

Line 154: The authors should consider to add photos when describing the protocol.

Thank you for pointing this out, we agree this is of value when describing the protocol and have added photos of exercises to Appendix B.

Line 372: If only one dog was in conservative treatment, the authors should consider to exclude this case. 

This has been discussed in the process by the authors and referring to the answer from question “line 125” above, we wanted to study the effect of therapeutic exercise for various stifle injuries with our chosen outcome measures being valid for measuring stifle injury and not specific for surgical cases, meaning that “having surgery” was not an inclusion criterion. Interestingly, nearly all dogs underwent surgery, and this mirrors the way dogs with stifle injury are treated in the area, however, this could be different in other regions or countries. As mentioned in the discussion under “4.2. Methodological considerations and future studies”, there were no substantial differences in results after 12 weeks of rehabilitation for the dog going through conservative treatment, but larger studies are warranted including more conservative treated dogs.

We agree it would have been desirable to prolong the time of the study to recruit more cases and possibly more dogs with conservative treatment, however, this study originally being a master thesis limited the time of the study to only a few months.

In the discussion section the authors should discuss the early introduction of hydrotherapy and other possible exercises. Compare to other studies and bibliographic references.

We have made the following addition to the Discussion on page 13, line 323-334:   

“The veterinary physiotherapist has an essential role in applying rehabilitation principles and in the progression of therapeutic exercise. For example, hydrotherapy can be used in different stages of rehabilitation (55–57). Early introduction of underwater treadmill exercise (subacute/reparative phase), with slow speed and adjusting the depth of water according to preferred ROM, promotes limb loading and activation of hypotrophied muscles, while also improving proprioception and balance (55,56). The home exercise protocol described in Appendix B, as mentioned earlier, was developed to address the six systems of balance control (Figure 1), as described by Witter and Bockstahler (58) and Millis and Levine (59) they are used to promote limb loading, activate muscles, and reduce pain by promoting neuromuscular interactions. While following the four principles of rehabilitation, adapted to the patient’s current status, we believe the exercises are safe for early introduction.”

Is this a study in a clinical setting? If so, the authors should state this.

This study is a randomized controlled clinical trial as stated in section “2.1. Study design”. To clarify the study took place in a clinical setting we have rephrased a sentence in section “2.3. Procedure”, page 7, line 160-162, as follows:

“Rehabilitation sessions were part of the dog’s veterinary treatment plan, in consultation with a team of animal health care professionals together with the dog owners, in a clinical setting.”

Reviewer 3 Report

Comments and Suggestions for Authors

Thank you for submitting this interesting work on the impact of therapeutic exercise on body-weight distribution, balance, and stifle function in dogs affected with different stifle injury conditions, most of them submitted to surgery, with an impact on the research area of canine orthopedic rehabilitation. In this randomized controlled trial, the authors evaluated the effects of a 12-week progressive therapeutic home exercise protocol on three-legged standing, targeting balance, postural-, and neuromuscular control and disability in dogs with stifle injury. The progressive therapeutic exercise protocol improved to a larger extent the static body-weight distribution between hind limbs, pain-related functional disability, and stifle function.

Lines 282, 283. Regarding the sentence in the discussion session: ‘This study is a randomized controlled trial and to our knowledge the first to evaluate the effect of therapeutic exercise as an add-on intervention‘. I believe there are already a few studies in this area, as the authors mention in the abstract: ‘Earlier studies have reported the benefits of rehabilitation after stifle injury’ and that it would be better to reformulate this sentence.

There are 66 references. Some may not be necessary. Please revise for example 'Johnson JA, Austin C, Breur GJ. Incidence of canine appendicular musculoskeletal disorders in 16 veterinary teaching hospitals 510 from 1980 through 1989. Vet Comp Orthop Traumatol. 1994;7(02):56–69. DOI: 10.1055/s-0038-1633097' for adequacy.

Comments on the Quality of English Language

Some spelling issues may be improved: For example, use: 'Both groups improved after the intervention period, but the group using a progressive therapeutic exercise protocol improved to a larger extent regarding static body-weight distribution between hindlimbs.' instead of 'Both groups improved after the intervention period, but the progressive therapeutic exercise protocol improved to a larger extent regarding static body-weight distribution between hindlimbs'.

Author Response

Authors response to reviewer 3

Lines 282, 283. Regarding the sentence in the discussion session: ‘This study is a randomized controlled trial and to our knowledge the first to evaluate the effect of therapeutic exercise as an add-on intervention‘. I believe there are already a few studies in this area, as the authors mention in the abstract: ‘Earlier studies have reported the benefits of rehabilitation after stifle injury’ and that it would be better to reformulate this sentence.

Thank you for pointing out that this is not correctly phrased, we have rephrased it on page 13, line 293-295, as follows:

“This study is a randomized controlled trial and to our knowledge one of the few experimental studies to evaluate the effect of therapeutic exercise as an add-on intervention on balance, function, and static body-weight distribution.”

There are 66 references. Some may not be necessary. Please revise for example 'Johnson JA, Austin C, Breur GJ. Incidence of canine appendicular musculoskeletal disorders in 16 veterinary teaching hospitals 510 from 1980 through 1989. Vet Comp Orthop Traumatol. 1994;7(02):56–69. DOI: 10.1055/s-0038-1633097' for adequacy.

Thank you for highlighting this, we have revised and decided to remove references 4 and 13 not being necessary to this study.

4. Johnson JA, Austin C, Breur GJ. Incidence of canine appendicular musculoskeletal disorders in 16 veterinary teaching hospitals from 1980 through 1989. Vet Comp Orthop Traumatol. 1994;7(02):56–69. DOI: 10.1055/s-0038-1633097

13. O’Connor BL, Visco DM, Heck DA, Myers SL, Brandt KD. Gait alterations in dogs after transection of the anterior cruciate ligament. Arthritis Rheum. 1989 Sep;32(9):1142–7. DOI: https://doi.org/10.1002/anr.1780320913

Comments on the Quality of English Language

Some spelling issues may be improved: For example, use: 'Both groups improved after the intervention period, but the group using a progressive therapeutic exercise protocol improved to a larger extent regarding static body-weight distribution between hindlimbs.' instead of 'Both groups improved after the intervention period, but the progressive therapeutic exercise protocol improved to a larger extent regarding static body-weight distribution between hindlimbs'.

We have now made another spelling check of the manuscript.

Round 2

Reviewer 1 Report

Comments and Suggestions for Authors

Dear authors, thanks you for the revision. The paper is now ready for publication. Congrats!